# Molecular Epidemiology of *Blastocystis* in Confined Slow Lorises, Macaques, and Gibbons

**DOI:** 10.3390/ani12212992

**Published:** 2022-10-31

**Authors:** Qingyong Ni, Shasha Dong, Yumai Fan, Wen Wan, Ping Teng, Shaobo Zhu, Xiaobi Liang, Huailiang Xu, Yongfang Yao, Mingwang Zhang, Meng Xie

**Affiliations:** 1Farm Animal Genetic Resources Exploration and Innovation Key Laboratory of Sichuan Province, Sichuan Agricultural University, Chengdu 611134, China; 2Key Laboratory of Livestock and Poultry Multi-Omics, Ministry of Agriculture and Rural Affairs, College of Animal Science and Technology, Sichuan Agricultural University, Chengdu 611134, China; 3Dehong Wildlife Rescue Center, Forestry Bureau of Dehong Prefecture, Mangshi 678499, China; 4College of Life Science, Sichuan Agricultural University, Ya’an 625099, China

**Keywords:** *Blastocystis* sp., NHPs, subtypes, captive management, zoonotic potential

## Abstract

**Simple Summary:**

*Blastocystis* sp. is one of the most frequently reported parasites and an intestinal anaerobic organism with high genetic diversity, which seriously threatens the health of non-human primates (NHPs). We conducted a comparative study based on small subunit ribosomal RNA (SSU rRNA) to assess the *Blastocystis* infection, its gene subtypes, and its zoonotic potential in confined macaques, gibbons, and slow lorises. The results indicated that the *Blastocystis* infections significantly differed between seasons and species. This is the first report on the prevalence and subtype distribution of *Blastocystis* sp. among captive slow lorises (*Nycticebus* spp.). The findings may contribute to formulating a feasible strategy for the improved confined management of NHPs and subsequent conservation practices, such as soft release.

**Abstract:**

*Blastocystis* sp. is a common intestinal anaerobic parasite infecting non-human primates and many other animals. This taxon threatens the health of NHPs due to its high genetic diversity, impeding efforts to improve confined management and subsequent conservation practices. This study collected 100 and 154 fecal samples from captive macaques, gibbons, and slow lorises in the summer and winter, respectively. The *Blastocystis* infection, its gene subtypes, and its zoonotic potential based on small subunit ribosomal RNA (SSU rRNA) were analyzed. The prevalence of *Blastocystis* in the three primate genera was 57.79% (89/154) in the summer (2021) and 29.00% (29/100) in the winter (2020). Four zoonotic subtypes—ST1, ST2, ST3, and ST4—were identified. ST2 was the most prevalent subtype, suggesting that these animals may serve as reservoirs for pathogens of human *Blastocystis* infections. The macaques showed a more significant variation in *Blastocystis* infection between seasons than gibbons and slow lorises. The slow lorises in small cages and enclosure areas were potentially more infected by *Blastocystis* in the summer, indicating that inappropriate captive management may have detrimental effects on their health.

## 1. Introduction

Parasites are some of the most common exogenous pathogens in humans and non-human primates (NHPs) [1,2]. They may cause tissue damage or compression injuries to the host, depleting the host’s nutrients and thus propagating toxins or transmitting diseases [3]. Studies suggest that parasites affect the adaptability of the host and the survival and reproduction of infected individuals, posing a severe threat to the health of captive and wild primates [4]. Endoparasites, such as *Capillaria* sp. and *Entamoeba* sp., have been reported in NHPs and other animals [5,6,7]. In particular, primate taxa have been widely reported to be infected by the common luminal intestinal parasitic protist *Blastocystis* [1,5,8].

*Blastocystis* sp. is an intestinal protozoan with a broad host spectrum that mainly colonizes the ileocecal region of the host [2]. *Blastocystis* infections have been widely reported in humans and domestic, companion, and confined wild animals [9,10,11]. According to the survey data on *Blastocystis* infections in China, the prevalence in various populations is 0.007~48.6% [8]. The infected animals exhibit diarrhea, abdominal pain, nausea, and gastric distension [2,9,12,13]. Previous studies have shown that *Blastocystis* is related to irritable bowel syndrome (IBS) and inflammatory bowel disease (IBD) in human and animals, while it interreacts with other parasites such as *Giardia duodenalis* [12,14,15]. The prevalence of *Blastocystis* sp. varies greatly in NHPs [5,16,17,18,19,20]. Studies postulate that zoonotic transmission from confined wild animals may cause human *Blastocystis* infections [21,22]. The infection can be self-limiting under normal immune function [23]. Studies have shown that iodoquinol, paromomycin, and nitazoxanide (IPN) have a strong therapeutic effect on *Blastocystis hominis,* but it the infection is usually treated with a single drug such as metronidazole [24,25]. A *Blastocystis* sp. infection is a foodborne zoonosis attributed to eating food contaminated with pathogenic microorganisms [26]. Recently, the prevalence, host range, and subtype distribution of *Blastocystis* in humans and domestic animals, and the factors related to the transmission of zoonoses, have been widely recorded [16,27,28,29].

To date, 32 subtypes of *Blastocystis* have been proposed based on SSU rRNA gene sequencing [30]. However, only 28 subtypes (ST1–ST17, ST21, and ST23–ST32) are accepted due to their unabridged sequence data and meeting the current criteria for unique subtype designations. ST1–ST4 are the predominant STs, suggesting zoonotic potential [31]. ST1–ST9 and ST12 often appear in human *Blastocystis*-like infection cases [32]. Other subtypes are only found in specific animals, indicating a certain degree of recessive host specificity [1,31]. For example, hoofed animals, such as pigs, are the natural hosts of ST5 [10]. ST4 is the most common subtype in rodents [32], while ST6 and ST7 are common in birds [33]. Previous studies suggest that the most common subtypes in NHPs are ST1, ST2, ST3, and, especially, ST2 [34,35] (Table 1).

Macaques, gibbons, and slow lorises are the main primate taxa distributed in China, and all their species are listed as protected animals [36]. Many individuals were confiscated or rescued by wildlife zoos and rescue centers, and most of them were finally released into the wild. However, a hard release without an investigation of parasite infections may pose a severe threat to the local wild populations and lead to an unsuccessful reintroduction. Previous studies showed that the prevalence of *Blastocystis* subtypes varied in some of these species (Table 1). For example, macaques are mostly infected with ST1, ST2, and ST3 [33,35], while gibbons are infected mainly by ST1, ST2, ST3, ST5, and ST8 [1,37,38]. However, there is still a data deficiency regarding strepsirrhines such as *Nycticebus* spp. [38]. We conducted a comparative study on the prevalence and phylogeny of *Blastocystis* in the three primate genera (*Macaca, Hylobates*, and *Nycticebus*). The confined animals may encounter health-threatening situations, such as incorrect diets, disease, disability, fear, pain, and feeding environments inconsistent with their natural survival [39]. This investigation into parasitic infections can provide a reference for these animals’ health statuses and contribute to designing appropriate captive management schemes and improving animal welfare.

**Table 1 animals-12-02992-t001:** The prevalence and subtypes of *Blastocystis* reported in macaques and gibbons.

Host	Number of Positive/Total Samples	*Blastocystis* Subtypes					References
		ST1	ST2	ST3	ST8	ST19	
Macaques							
*Macaca mulatta*	6/18		4	2			[13]
*Macaca fascicularis*	3/13		2	1			[13]
*Macaca fuscata*	6/33		2	4			[13]
*Macaca mulatta*	28/29	15	5	7		1	[18]
*Macaca mulatta*	149/323	85	75	140			[20]
*Macaca nemestrina*	36/97	24	9	36			[20]
*Macaca fascicularis*	50/85	39	26	50			[20]
*Macaca nigra*	1/7	1					[35]
*Macaca nigra*	2/4	2					[21]
*Macaca tonkeana*	2/2	2					[21]
*Macaca mulatta*	12/30	10			2		[40]
Gibbons							
*Hylobates* spp.	2/18	1	1				[35]
*Nomascus leucogenys*	1/4	1					[13]
*Hylobates moloch*	2/4	2					[20]
*Hylobates leucogenys*	2/4	2					[20]
*Nomascus leucogenys*	2/2	2					[41]

## 2. Materials and Methods

### 2.1. Experimental Ethics

Sample collection and experimental protocols were approved by the Institutional Review Board (IRB13627), the Institutional Animal Care and Use Committee of the Sichuan Agricultural University, China, under permit number DKY-2020302166, and by the Administration for Wild Animal Protection in Yunnan Province, China. The protocols adhered to the American Society of Primatologists’ Principles for the Ethical Treatment of Non-Human Primates. Fecal samples were collected from NHPs per the Animal Ethics Procedures and Guidelines of the People’s Republic of China. 

### 2.2. Sample Collection and Recording

The study was conducted at the Dehong Wildlife Rescue Center in Yunnan, China (24.38287° N, 98.45872° E), between November 2020 and July 2021. Most of the animals were rescued after confiscation due to illegal hunting and trade. Fecal samples were collected from the trays placed under the cages. We cleared the trays at 1 a.m. and collected the feces at 6 a.m. to ensure that all the samples were collected within 5 h after defecation. A total of 254 fecal samples from five non-human primate species (*N. bengalensis, H. hoolock*, *M. leonina*, *M. arctoides*, and *M. mulatta*) were collected in summer (*n* = 154) and winter (*n* = 100) (Table 2). The macaques were kept with 1–3 individuals in each cage (90 × 100 × 90 cm), while the gibbons were housed with one or two individuals in a large cell (1000 × 500 × 600 cm). The Bengal slow lorises were grouped into two confined enclosures (A and B) and individually housed in two types of iron cages: small (30 × 20 × 20 cm) and larger cages (40 × 30 × 30 cm). The animals that had been housed longer than 12 months were defined as group LC (long-term confinement), while those confiscated within 12 months were defined as group SC (short-term confinement) (Table 3). The collected fecal samples were maintained in dry ice, transferred to the laboratory within 12 h, and stored at −80 °C until used. We also obtained the individual records (sex, captivity duration, and bodyweight) from the rescue center. However, the documents were unavailable in the winter due to the absence of technicians during the long-term pandemic lockdown. Only the data collected in the summer were used in this study. 

### 2.3. DNA Extraction, PCR Amplification, and Sequencing

DNA extraction was performed using Tiangen Tianamp stool DNA kit stool genomic DNA Extraction Kit (Catalog No. DP328, version No. DP201101x), following the manufacturer’s instructions. About 180–220 g of stool was placed into a 2 mL sterilized centrifuge tube using a sterilized toothpick and put on ice. The extracted DNA was stored at −20 °C until PCR amplification. The PCR mixture (25 μL) contained 12.5 μL of Taq PCR Master Mix (Tsingke Biotech Co., Ltd., Beijing, China), 1 μL of each primer (0.4 μM), 2 μL of genomic DNA, and nuclease-free water up to the desired volume. The PCR was started at 94 °C for 4 min, followed by 35 cycles of 95 °C for 45 s, 49 °C for 45 s, and 72 °C for 50 s, with an extension at 72 °C for 5 min. *Blastocystis* sp. was screened and subtyped through PCR amplification of an approximately 600 bp region of the SSU rRNA gene using the forward primer RD5: (5′-GGAGGTAGTGACAATAAATC-3′) and reverse primer BhRDr: (5′-TGCTTTCGCACTTGTTCATC-3′) [42,43]. One μL of each PCR product was electrophoresed on 1.5% agarose gel. Amplicons were stained with 0.5μg/mL of ethidium bromide and visualized using a UV Transilluminator. All PCR products were sequenced using the sanger method on an ABI 3730 sequencer (Bioneer, Daejeon, South Korea). 

### 2.4. Sequence and Phylogenetic Analysis

The nucleotide sequences generated in the current study were manually edited using SeqMan in DNASTAR v7.1 (https://www.dnastar.com/ accessed on 13 March 2022). All the sequences were aligned using SeqMan and reference sequences of known *Blastocystis* subtypes retrieved from NCBI GenBank (https://www.ncbi.nlm.nih.gov/ accessed on 8 March 2022). BLAST searches in the GenBank database (https://blast.ncbi.nlm.nih.gov/Blast.cgi accessed on 1 October 2021.) were conducted for all the sequences (excluding vector and primer sites) to confirm they were *Blastocystis* sp. and check their subtype [44]. We used the ClustalW algorithm in the MegAlign module (DNASTAR Inc., Madison, WI, USA) to align the nucleotide sequences generated in this study with type sequences from GenBank, representing the ten subtypes currently recognized. The aligned sequences were manually edited to remove any artificial gaps. The calculated evolutionary distances were used to analyze phylogenetic relationships to support subtype grouping by both the neighbor-joining (NJ) and Maximum Likelihood (ML) methods using MEGA 6.06. A phylogram was reconstructed using the Kimura 2-parameter model with 1000 bootstrap replicates (http://www.Mega.software.net/ accessed on 9 June 2022). This study’s unique partial SSU rRNA genes were deposited in GenBank under accession numbers ON714609–ON714630.

### 2.5. Statistical Analysis 

Statistical analyses were performed using the Statistical Package for the Social Sciences (SPSS) software version 27.0. We analyzed the differences between seasons and species using the Chi-square and fisher’s exact tests. We calculated 95% confidence intervals (CI) and the odds ratios (ORs) to determine the variation in infectivity rates and the effects of captive management factors. ORs were calculated using a two-by-two table, and ORs greater than 1.0 indicated a positive association between infections and the related factors. The significance threshold was set at *p* < 0.05. 

## 3. Results

### 3.1. Prevalence of Blastocystis sp. in NHPs

The prevalence of *Blastocystis* sp. in the three primate genera in the Dehong wildlife rescue center is shown in Table 2. The average prevalence in the winter was 29.00% (29/100). *Macaca* spp. had the highest prevalence (25/26; 96.15%), followed by *Hylobates* spp. (1/2; 50.00%), and *Nycticebus* spp. (3/72; 4.17%). The average prevalence in the summer was 57.79% (89/154). The macaques had a higher infection rate (54/56; 96.43%), followed by gibbons (7/13; 53.85%) and slow lorises (28/85; 32.94%). The two species of macaques, *M. arctoide* (*n* = 2) and *M. leonine* (*n* = 3), were all infected with *Blastocystis*.

### 3.2. Characterization of Blastocystis sp. Subtypes

Table 2 shows the infection prevalence and subtypes of *Blastocystis* in the five primate species. The gene subtypes of *Blastocystis* protozoa during the winter in *Nycticebus* spp. were ST1(1/3), ST3(1/3), and ST4(1/3). The macaques were mainly infected with ST2(18/25), while gibbons were infected with ST1(1/1). The prevalent gene subtypes of *Blastocystis* during the summer were ST2(25/28) and ST2(39/54) in *Nycticebus* spp. and *Macaca* spp., respectively. All positive *Hylobates* spp. were infected with ST2(7/7). 

### 3.3. Blastocystis sp. Infection in Nycticebus spp.

The infection rates between species differed significantly during the summer (Chi-square test: *x*^2^ = 55.874; *p* < 0.05) and winter (Fisher’s exact test; *p* < 0.05). The infection rates showed a significant difference between the winter and summer in macaques (*x*^2^ = 20.367; *p* < 0.05), but no significant difference occurred in gibbons and slow lorises. The Chi-square test was also used to analyze the relationship between infected and uninfected groups of slow lorises and the correlations with confinement duration, sex, enclosure area, cage size, and bodyweight during the summer (Table 3). The results showed a significant difference in the infection rates among individuals housed in different enclosures and cage sizes. Based on the OR value, there was a positive correlation between the captivity duration and the infection rate of slow lorises. Long-term confinement may increase the infection rate of *Blastocystis.* The cage and enclosure size are also strongly correlated with the infection rate. The chi-square test and odds ratio results showed that sex and bodyweight had no significant effects on the infection rate of *Blastocystis*.

### 3.4. Phylogeny of Blastocystis sp.

A total of 22 representative sequences were selected from 118 positive *Blastocystis* sp. isolates and were used to construct a phylogenetic tree (Figure 1). The sequences obtained in this study showed identity with nine reference sequences of *Blastocystis* sp. published in GenBank. The newly acquired sequences belonged to ST1, ST2, ST3, and ST4. ST1–ST3 were found in the three primate taxa, and the sequences clustered together. ST4 was only detected in slow lorises and clustered with human-derived sequences (Figure 1).

## 4. Discussion

The difference in the prevalence of *Blastocystis* sp. in the summer and winter was significant in the three species, indicating that the extent of parasitic infection varied significantly between seasons. Previous studies have demonstrated that the prevalence of some dominant niches of protozoa varied in different hydrological periods [45]. For example, in the wet season, *Giardia* sp. was more often detected in free-ranging black howler monkeys (*Alouatta pigra*) than in the dry season [46]. The parasites may also have higher infection rates in the summer than in other seasons [47]. Higher temperatures and precipitation in the summer and rainy seasons may provide favorable conditions for endoparasites’ growth [47,48]. 

The slow lorises and gibbons displayed lower infection rates of *Blastocystis* than macaques in this study. The significant differences between the three primate taxa were potentially consistent with the susceptibility and recessive host specificity toward *Blastocystis* [9]. Some studies suggest that the venom of slow lorises has a specific defensive effect against ectoparasites [49,50], but there is no direct evidence for its function in defending against endoparasites. Human interference and the threat of illegal animal trade constitute an inevitable intersection between wildlife populations, livestock, and human beings, facilitating the cross-species transmission of potential pathogens [4]. Through the sharing of poor husbandry conditions without regular veterinary follow-ups, food or waterborne parasites could spread in humans and animals. In addition, survival stresses, such as a lack of food supply, water, and living space, increase the risk of parasitic infections in animals [51]. Compared with the macaques and slow lorises, which are housed together in tiny cages, the gibbons in the Dehong wildlife rescue center are kept alone or in pairs within large enclosures (300 m^3^), possibly leading to a lower infection rate of *Blastocystis* protozoa. 

Macaques were the most susceptible to *Blastocystis* among the three taxa. This can be explained as a direct consequence of the fact that the macaques fed directly from the ground and drank water with the same ladle. Other studies postulate that macaques can be infected with various parasites [31,33,35]. The results may be due to the common confinement in groups since animals have a higher infection rate when mixed with others, especially at different ages [51,52]. Parasite transmission rates may scale positively with the host population’s density or the frequency of interactions [53]. Macaques have a larger body size, home range, and dietary width in the wild than the small-sized slow lorises [51]. Restricting their nutritional and habitat supply during confinement makes them susceptible to parasitic infections.

*Blastocystis* demonstrates extensive host adaptability and specificity at the gene subtype level [11]. The four subtypes (ST1, ST2, ST3, and ST4) detected in this study are all zoonotic subtypes [12,31], suggesting that zoonosis could have occurred in Dehong wildlife rescue center, which remains to be demonstrated. ST1, ST2, and ST3 were the prevalent subtypes in humans and NHPs. ST2 was the most pervasive subtype [2,34,35], while ST3 was the most virulent subtype among the 32 subtypes [43]. ST4 is not commonly found in NHPs [54]. Only the slow lorises were infected with ST4, possibly due to host specificity. *Blastocystis* ST4 may have beneficial effects on intestinal commensal bacteria in vitro and can inhibit the growth of pathogenic *B. vulgatus* [55].

Due to the absence of data in the winter, the effects of the confinement duration, sex, enclosure area, cage size, and body weight on the prevalence and subtype distribution of *Blastocystis* in *N. bengalensis* were only examined in the summer. More individuals in the SC group appeared to be infected by *Blastocystis* than those in the LC group. We speculate that the SC individuals, especially newly rescued ones, may still carry parasites from the wild. In addition, the SC individuals were housed in the enclosure close to the macaques, leading to a high risk of pathogenic transmission. Similarly, survival stresses, such as a lack of food supply and home range, make slow lorises more vulnerable to parasites and infectious diseases [4,51], especially with respect to those in small cages. It is highly recommended to provide large enclosures and separate feed boxes for the captive macaques and slow lorises and accelerate release progress for the confiscated individuals. We also recommend that sick individuals be provided with separate rooms and treatments. Individual characteristics, such as sex and body weight, have been reported to be correlated with parasitic infection. To be specific, sex affects the response of intestinal microorganisms to the host’s parasites, and weight loss may result from the reaction of the immune system to parasites [56,57]. We found that sex and body weight had no significant correlations with *Blastocystis* infection in slow lorises. However, the findings should be confirmed in further studies. 

## 5. Conclusions

Four zoonotic subtypes infected five species in three primate genera, indicating the possible transmission between wild primates and humans in rescue centers. This study is the first systematic report on *Blastocystis* sp. infections in *Nycticebus* spp. The results suggest a higher host specificity towards ST4 in slow lorises than in macaques and gibbons. Captive management played a substantial role in affecting the *Blastocystis* infections of the confined NHPs. To alleviate detrimental effects, rescue centers or zoos are highly recommended to provide larger enclosures and cages for the slow lorises and macaques and house them separately. Given the absence of data in the winter, further research is needed to confirm the findings regarding seasonal variation.

## Figures and Tables

**Figure 1 animals-12-02992-f001:**
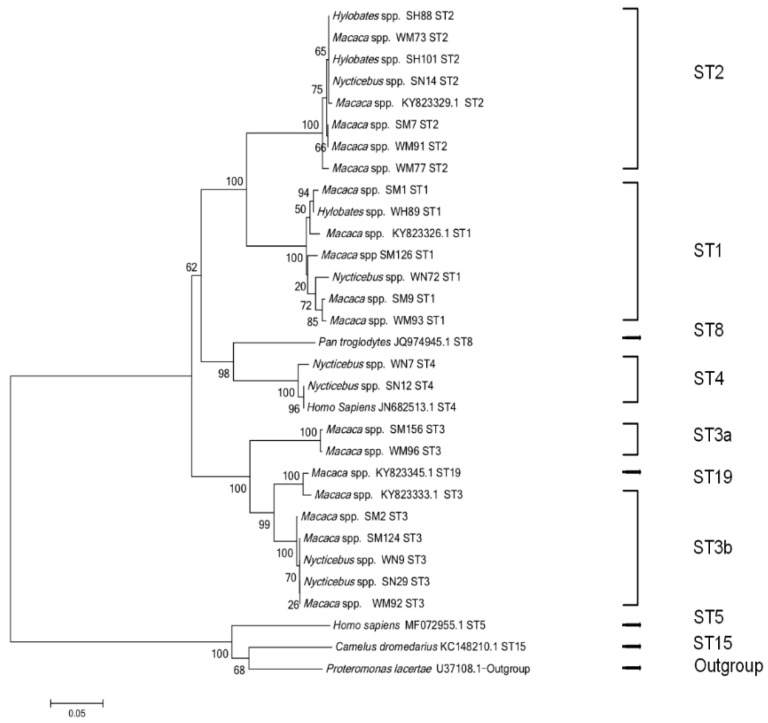
Phylogenetic relationship of *Blastocystis* sp. based on the nucleotide sequences of the barcode regions of small subunit ribosomal RNA (SSU rRNA). The neighbor-joining method was used to construct the trees using the Kimura-2-parameter model. The numbers on the branches are the percent bootstrap values from 1000 replicates. Each sequence was identified by its host, accession/association number, and subtype.

**Table 2 animals-12-02992-t002:** Prevalence and subtype distribution of *Blastocystis* in NHPs.

Species	Prevalence (95%CI)		Subtypes (Number)	
	Summer	Winter	Summer	Winter
Slow lorises				
*Nycticebus bengalensis*	28/85, 32.94% (22.95~42.93%)	3/72, 4.17% (0~8.78%)	ST2 (25), ST3 (2), ST4 (1)	ST1 (1), ST3 (1), ST4 (1)
Macaques				
*Macaca mulatta*	51/53, 96.23% (91.10~100%)	20/21, 95.24% (86.13~100%)	ST1 (10), ST2 (38), ST3 (3)	ST1 (3), ST2 (14), ST3 (3)
*Macaca arctoides*	2/2, 100.00% (0~100%)	2/2, 100.00% (0~100%)	ST2 (1), ST3 (1)	ST1 (1), ST2 (1)
*Macaca leonina*	1/1, 100.00% (0~100%)	3/3, 100.00% (0~100%)	ST3 (1)	ST2 (3)
Gibbons				
*Hylobates Hoolock*	7/13, 53.85% (26.75~80.95%)	1/2, 50.00% (0~100%)	ST2 (7)	ST1 (1)

**Table 3 animals-12-02992-t003:** The Chi-square test and OR values showed the effects of confinement duration, sex, enclosure area, cage size, and bodyweight on the prevalence and subtype distribution of *Blastocystis* in *N. bengalensis* in summer.

	Infected (*n* = 28)	Uninfected (*n* = 57)	OR (95%CI)	*x* ^2^	*p*
Captive duration			2.976 (0.783~11.307)	2.738	0.098
LC	25	42			
SC	3	15			
Sex			0.837 (0.837~2.071)	0.149	0.700
Male	13	29			
Female	15	28			
Enclosure area			0.376 (0.147~0.959)	4.308	0.038
A	10	34			
B	18	23			
Cage size			4.512 (1.709~11.918)	9.848	0.002
Small	16	12			
Large	13	44			
Bodyweight			1.026 (0.389~2.707)	0.003	0.958
I (≥1 kg)	39	18			
II (<1 kg)	19	9			

LC—long-term confinement; SC—short-term confinement; OR—odds ratio.

## Data Availability

This study’s unique partial SSU rRNA genes were deposited in GenBank (https://www.ncbi.nlm.nih.gov/genbank/ accessed on 9 June 2022) under accession numbers ON714609-ON714630.

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
