# Peer review of "Molecular Epidemiology of Blastocystis in Confined Slow Lorises, Macaques, and Gibbons"

_animals, 2022, doi:10.3390/ani12212992_

Round 1
Reviewer 1 Report (New Reviewer)
My suggestions and comments to improve the manuscript are the following:
- Delete the following sentences: ‘Parasites are divided into ecto- and endo-parasites. Ectoparasites …. Macaca mulatta.’ The manuscript is focused in intestinal protozoan.
- ‘Blastocystis is an intestinal protozoan …’ Blastocystis is a taxonomic genus. The designation Blastocystis sp. is considered more appropriate.
- Write Giardia lamblia in italics; also Giardia duodenalis is the more recently accepted species name;
- The sentence ‘Many studies report that parasitic nematodes in primate communities infect each other based on the morphological description, thus misestimating the cross-infection between closely related species and inaccurately inferring the disease risk [22]’ does not make sense. Please explain and reorganize the information.
- ‘Blastocystis is a foodborne zoonosis attributed to eating food contaminated with pathogenic microorganisms [23].’ Blastocystis sp is a protozoan genus; Blastocystosis is the infection;
- Please explain why ‘PCR products were pooled and visualized by agarose gel electrophoresis.’ Before the PCR itself.
- Delete ‘To analyze the base sequence of specific DNA fragments-the arrangement of ATCG.’
- The sentence ‘The absence of intermediate hosts like mosquitoes during winter compared to summer may decrease the possible transmission routes, leading to a lower infection rate [43].’ Reveals little knowledge about Blastocystis life-cycle and parasitology. Please do not make such considerations.
- Please explain why ‘The parasites also have little or weak activity in autumn and winter.’ Many parasites, especially food and water borne, can occur in rainy seasons.
- Is there any evidence that slow lorises venom acts as deworming against protozoan parasites as stated in the following sentence: ‘The venom may also help defend against endopara-sites in the same way that secondary plant metabolites help treat bacterial and parasitic infections [47].’
- Just because some animals are infected with the zoonotic subtypes it doesn’t mean that the infection has infectied humans as suggested in the sentence ‘The four subtypes (ST1, ST2, ST3, and ST4) detected in this study are all zoonotic subtypes [11, 28], suggesting zoonosis occurred in Dehong wildlife rescue center.’
Author Response
To reviewer 1
- Delete the following sentences: ‘Parasites are divided into ecto- and endo-parasites. Ectoparasites …. Macaca mulatta.’ The manuscript is focused in intestinal protozoan.
Yes, I did.
- ‘Blastocystis is an intestinal protozoan …’ Blastocystis is a taxonomic genus. The designation Blastocystis is considered more appropriate.
Yes, I agree.
- Write Giardia lamblia in italics; also Giardia duodenalis is the more recently accepted species name;
Yes, I agree.
- The sentence ‘Many studies report that parasitic nematodes in primate communities infect each other based on the morphological description, thus misestimating the cross-infection between closely related species and inaccurately inferring the disease risk [22]’ does not make sense. Please explain and reorganize the information.
Thank you. This sentence is not consistent with the context. I deleted it.
- ‘Blastocystis is a foodborne zoonosis attributed to eating food contaminated with pathogenic microorganisms [23].’ Blastocystis sp is a protozoan genus; Blastocystosis is the infection;
Yes, I did it.
- Please explain why ‘PCR products were pooled and visualized by agarose gel electrophoresis.’ Before the PCR itself.
We reorganized this part.
- Delete ‘To analyze the base sequence of specific DNA fragments-the arrangement of ATCG.’
Yes, I did.
- The sentence ‘The absence of intermediate hosts like mosquitoes during winter compared to summer may decrease the possible transmission routes, leading to a lower infection rate [43].’ Reveals little knowledge about Blastocystis life-cycle and parasitology. Please do not make such considerations.
We reorganized this part.
- Please explain why ‘The parasites also have little or weak activity in autumn and winter.’ Many parasites, especially food and water borne, can occur in rainy seasons.
Thanks for your advice. We rephrased it.
- Is there any evidence that slow lorises venom acts as deworming against protozoan parasites as stated in the following sentence: ‘The venom may also help defend against endopara-sites in the same way that secondary plant metabolites help treat bacterial and parasitic infections [47].’
There is no direct evidence. We rephrased this sentence.
- Just because some animals are infected with the zoonotic subtypes it doesn’t mean that the infection has infectied humans as suggested in the sentence ‘The four subtypes (ST1, ST2, ST3, and ST4) detected in this study are all zoonotic subtypes [11, 28], suggesting zoonosis occurred in Dehong wildlife rescue center.’
We rephrased this sentence.

Reviewer 2 Report (New Reviewer)
Information on the prevalence of Blastocystis sp. in captive NHP is valuable to assist in strategies to improve the health and welfare of captive animals and also to prevent zoonotic transmission to animal keepers, volunteers, visitors, and nearby residents. This study outlined the prevalence of Blastocystis sp. in several NHP species of an animal sanctuary with mention of potential risk factors for transmission to and from the animals. Additional information regarding the exposure pathways for infection among the animals and their caregivers is needed to further explain the threat of the parasite in this unique environment and highlight opportunities for prevention and control efforts.
Simple Summary
· Spell out NHP here, too, so readers who start reading at the very beginning or in the introduction will each know the population of interest for this study
· Add slow loris either before or after Nycticebus spp. to notify unfamiliar readers to which NHP group this species belongs
Abstract
· After the text summer and winter, add the years of sample collection (ex. 2020-2021) so the reader knows which summer and winter the study took place
Introduction
· Explain further what is meant by “mechanical damage” or consider rewriting for clarity
· Have endoparasites been reported in most macaques (most of the macaques across the globe)? Or mostly in macaques, when considering the prevalence of these parasites among all NHP species?
· In the description of Blastocystis sp., add information about the forms of treatment for animals and humans and whether the disease can be self-limiting
· Giardia lamblia should be italicized
· When discussing Giardia sp. and IBS, is this infection and resulting symptoms in reference to animals or people? Or both?
· Throughout the paper, check capitalization of “subtype”
· When listing health-threatening situations for NHPs, move the terms ‘fear’ and ‘pain’ to before the last item, “…and feeding environments…”
· Table 1: Explain why there are several studies with the number of positive/total number of samples left blank in either a table notation or footnote
Materials and Methods
· Was fecal collection done by rectal sampling of the animals or by collecting feces deposited on the ground/in the cage?
· Why was information on the captive animals incomplete during the winter sample collection? Please provide a brief explanation.
· Clear description of the DNA extraction, PCR amplification, and sequencing process
Results
· Table 2: Consider changing the subheadings for each category of NHP to the broader terms previously used (ex. slow lorises, macaques, and gibbons) with the true species name listed below
· As the results described significant differences in infection rates among the slow lorises based on whether they were housed together and in small cages, this should be described more within the discussion section to address potential exposure pathways for infection (ex. is it coming from animal exposure from outside of their cage? Are the cages located inside a building? Or outdoors? Do they all have the same handlers/animal keepers? Are they housed separately when they are ill?)
Discussion
· How do mosquito vectors serve as a host for this gastrointestinal parasite? Explain and cite further and add this element of the transmission cycle to the introduction where it is described as a foodborne endoparasite.
· Missing detailed description about the seasonality of gastrointestinal zoonotic parasites, especially those that are food and waterborne, and how this may have influenced parasite prevalence in this population
· Need more explanation on how human interference and the illegal animal trade can increase the prevalence of parasite- describe how the transmission can occur for humans, animals, and their shared environment
· The four subtypes detected in the study only show the potential for zoonotic transmission, but in and of themselves cannot imply that zoonotic transmission has occurred unless the researchers examined and tested animal keepers, too
· Explain the high risk of pathogenic transmission between NHP species that have cages housed together. Need more information on the housing and care of the animals to know more about exposure risks
· When the authors express that the findings should be confirmed in further studies, do they mean in animals or in humans? Or both?
· What recommendations do the study authors have for Blastocystis sp. prevention in captive animal environments, especially in regard to housing, cage size, stress level, etc.?
Conclusion
· There is a great sentence here about rescue centers or zoos providing larger enclosures and cages and housing them separately, but the reviewer would like to see this addressed more in the discussion, especially in regard to how these factors could be contributing to the spread of the parasite between, and within, the NHP populations and to/from their human caregivers or guests
Author Response
To reviewer 2
Simple Summary
- 1 Spell out NHP here, too, so readers who start reading at the very beginning or in the introduction will each know the population of interest for this study
Yes, I did it.
- 2 Add slow loris either before or after Nycticebus spp. to notify unfamiliar readers to which NHP group this species belongs
Yes, I did it.
Abstract
- 3 After the text summer and winter, add the years of sample collection (ex. 2020-2021) so the reader knows which summer and winter the study took place
Yes, I did.
Introduction
- 4 Explain further what is meant by “mechanical damage” or consider rewriting for clarity
We rewrote this statement.
- 5 Have endoparasites been reported in most macaques (most of the macaques across the globe)? Or mostly in macaques, when considering the prevalence of these parasites among all NHP species?
This sentence is ambiguous. We rephrased it.
- 6 In the description of Blastocystis sp., add information about the forms of treatment for animals and humans and whether the disease can be self-limiting
We added the related statements in the second paragraph.
- 7 Giardia lamblia should be italicized
Yes, I did.
- 8 When discussing Giardia sp. and IBS, is this infection and resulting symptoms in reference to animals or people? Or both?
The symptoms may occur in both animals and people. We rephrased this statement.
- 9 Throughout the paper, check capitalization of “subtype”
We checked and corrected.
- 10 When listing health-threatening situations for NHPs, move the terms ‘fear’ and ‘pain’ to before the last item, “…and feeding environments…”
Yes, I did.
- 11 Table 1: Explain why there are several studies with the number of positive/total number of samples left blank in either a table notation or footnote
The data were not accessible. We replaced the data and related references
Materials and Methods
- 12 Was fecal collection done by rectal sampling of the animals or by collecting feces deposited on the ground/in the cage?
We placed trays under the cages and collected the fecal samples in five hours. We added this statement in “Sample collection and recording”.
- 13 Why was information on the captive animals incomplete during the winter sample collection? Please provide a brief explanation.
The records were unavailable in winter due to the absence of technicians during a long-term epidemic lockdown. We added the statement.
- 14 Clear description of the DNA extraction, PCR amplification, and sequencing process
Yes, I did.
Results
- 15 Table 2: Consider changing the subheadings for each category of NHP to the broader terms previously used (ex. slow lorises, macaques, and gibbons) with the true species name listed below
Yes, I did.
- 16 As the results described significant differences in infection rates among the slow lorises based on whether they were housed together and in small cages, this should be described more within the discussion section to address potential exposure pathways for infection (ex. is it coming from animal exposure from outside of their cage? Are the cages located inside a building? Or outdoors? Do they all have the same handlers/animal keepers? Are they housed separately when they are ill?)
We rephrased this part.
Discussion
- 17 How do mosquito vectors serve as a host for this gastrointestinal parasite? Explain and cite further and add this element of the transmission cycle to the introduction where it is described as a foodborne endoparasite.
We rephrased this part.
- 18 Missing detailed description about the seasonality of gastrointestinal zoonotic parasites, especially those that are food and waterborne, and how this may have influenced parasite prevalence in this population
We rephrased this paragraph.
- 19 Need more explanation on how human interference and the illegal animal trade can increase the prevalence of parasite- describe how the transmission can occur for humans, animals, and their shared environment
We added the related statements.
- 20 The four subtypes detected in the study only show the potential for zoonotic transmission, but in and of themselves cannot imply that zoonotic transmission has occurred unless the researchers examined and tested animal keepers, too
We rephrased this sentence.
- 21 Explain the high risk of pathogenic transmission between NHP species that have cages housed together. Need more information on the housing and care of the animals to know more about exposure risks
We added the related statements.
- 22 When the authors express that the findings should be confirmed in further studies, do they mean in animals or in humans? Or both?
We suggest that the findings in slow lorises need to be confirmed.
- 23 What recommendations do the study authors have for Blastocystis sp. prevention in captive animal environments, especially in regard to housing, cage size, stress level, etc.?
We added the related statements.
Conclusion
- 24 There is a great sentence here about rescue centers or zoos providing larger enclosures and cages and housing them separately, but the reviewer would like to see this addressed more in the discussion, especially in regard to how these factors could be contributing to the spread of the parasite between, and within, the NHP populations and to/from their human caregivers or guests
Thank you. We added some related statements in the discussion.

Reviewer 3 Report (New Reviewer)
The study is of interest and well designed and presented.
It is proposed that you revise once again the manuscript in terms of English and some other minor comments such as all parasite names in italics.
In the manuscript it is speculated that the absence of intermediate hosts like mosquitoes during winter compared to summer may decrease the possible transmission routes, leading to a lower infection rate. Blastocyst sp. are transmitted via the focal-oral route and mosquitoes are not involved. Perhaps an explanation lies in the fact that climate conditions depending on seasons affect the survival of the cysts. In my opinion you should take a closer look at this point and explain it regarding protozoan parasites.
In this life cycle context, the fact that the macaques were more susceptible to infection can be explained as a direct consequence of the fact that they fed directly from the ground and drank water with the same ladle.
Finally you could add in the discussion section some extra relevant references such as the paper entitled "Intestinal Protists in Captive Non-human Primates and Their Handlers in Six European Zoological Gardens. Molecular Evidence of Zoonotic Transmission" https://doi.org/10.3389/fvets.2021.819887
Author Response
To reviewer 3
The study is of interest and well designed and presented.
- It is proposed that you revise once again the manuscript in terms of English and some other minor comments such as all parasite names in italics.
We conducted an English polishing and checked/corrected the parasite names.
- In the manuscript it is speculated that the absence of intermediate hosts like mosquitoes during winter compared to summer may decrease the possible transmission routes, leading to a lower infection rate. Blastocyst sp. are transmitted via the focal-oral route and mosquitoes are not involved. Perhaps an explanation lies in the fact that climate conditions depending on seasons affect the survival of the cysts. In my opinion you should take a closer look at this point and explain it regarding protozoan parasites.
Thank you. We rephrased this paragraph.
- In this life cycle context, the fact that the macaques were more susceptible to infection can be explained as a direct consequence of the fact that they fed directly from the ground and drank water with the same ladle.
Thank you. We reorganized this part.
- Finally you could add in the discussion section some extra relevant references such as the paper entitled "Intestinal Protists in Captive Non-human Primates and Their Handlers in Six European Zoological Gardens. Molecular Evidence of Zoonotic Transmission" https://doi.org/10.3389/fvets.2021.819887
I added some related references.

Round 2
Reviewer 2 Report (New Reviewer)
The authors completed a quick and thorough revision to strengthen the findings and presentation of this study. Well done and best wishes.
This manuscript is a resubmission of an earlier submission. The following is a list of the peer review reports and author responses from that submission.
Round 1
Reviewer 1 Report
The manuscript by Ni et al. is well written and the results are interesting. Although I think it could be improved in some sections.
1. In the introduction it is better that the authors highlight some clinical importance of Blastocystis for humans, especially for zoonotic subtypes, and some important diseases such as IBS ns IBD.
You can use following papers:
· https://www.sciencedirect.com/science/article/abs/pii/S0882401021004897
· https://link.springer.com/article/10.1007/s10096-017-3065-x
2. For overall prevalence rates in the result section please provide 95%CI. Moreover, please present related methods, in the method section.
Reviewer 2 Report
Page 4, Line 121: Were the subtypes of Blastocystis identified by PCR or by sequencing?
Page 4, Line 127: Briefly explain the protocol for nucleotide sequencing
Page 6, Line 178: Was the association between infection rate and other factors (enclosure, cage size, confinement duration, body weight, and sex) also studied in winter?
Page 6, Table 3: Add "in summer" at the end of the title of Table 3 (since the associations between infectivity rate and other factors were performed only in summer)
Conclusions: Since most of the associations between infectivity rate and other factors such as enclosure, cage size, confinement duration, body weight, and sex etc. were performed only in summer due to lack of the data in winter, this should be mentioned in the conclusions. Also, mention that studying the associations in winter will further strengthen the findings of the study.